# A Simple and Reliable Liquid Chromatographic Method for Simultaneous Determination of Five Benzodiazepine Drugs in Human Plasma

**Hassan M. Albishri [1,\*], Naflaa A. Aldawsari [1] and Deia Abd El-Hady [2]**

[1] Chemistry Department, Faculty of Science, King Abdulaziz University, Jeddah 80203, Saudi Arabia; naldawsari0011@stu.kau.edu.sa

[2] Department of Chemistry, College of Science, University of Jeddah, Osfan Road, P.O. Box 80327, Jeddah 21589, Saudi Arabia; damoustafa@uj.edu.sa

\* Correspondence: hmalbeshri@kau.edu.sa; Tel.: +966-54-5550373

**Abstract:** Benzodiazepines (BZDs) are one of the most important drugs that have been used in the treatment of neuropsychological disorders. Indeed, BZDs are abused by drug addicts regardless of their therapeutic uses. Therefore, it was important in forensic and clinical toxicology to reach an easy and reliable method for the screening and quantification of BZDs in the human plasma matrix. In the current work, five BZDs, namely bromazepam, clonazepam, lorazepam, nordiazepam and diazepam were simultaneously separated and detected by a simple and reliable RPLC method in a human plasma matrix. Isocratic mobile elution consisting of 20 mmol $L^{-1}$ phosphate buffer (pH 7.0) and methanol (50:50, $v/v$) on a Symmetry C18 column was employed. The flow rate, wavelength and column temperature were fixed at 1.0 mL $min^{-1}$, 214 nm and 40 °C, respectively. The proposed method was validated, giving a linearity within the concentration ranges 5–500 ng $mL^{-1}$ for bromazepam and diazepam, 3–500 ng $mL^{-1}$ for clonazepam and lorazepam and 1–500 ng $mL^{-1}$ for nordiazepam with a determination coefficient ($R^2$) more than 0.9992. The LOD values for the selected BZDs ranged from 0.54 to 2.32 and from 1.78 to 7.65 ng $mL^{-1}$ for standard methanolic and plasma matrices, respectively. Precision, accuracy, selectivity, stability, and robustness were some of the terms considered in validating the current RPLC method. Based on these results, a simple and reliable RPLC method was successfully applied to quantify BZDs in human plasma matrix appearing with recoveries ranging from 96.5 to 107.5% and interday RSD less than 4%. The current developed method was useful for rapidly screening the most commonly used BZDs in the market within their therapeutic concentration ranges.

**Keywords:** benzodiazepines; reversed-phase liquid chromatography; plasma

## 1. Introduction

Psychiatric disorders attract the attention of public communities because they account for 22.8% of the global burden of diseases [1]. Among the main causes of this disability is depression, which has increased significantly since 1990 as a result of overpopulation and aging. Benzodiazepines (BZDs) are considered one of the types of narcotic drugs that slow down the functions of the body and the brain, which led to their use in the treatment of some diseases such as anxiety, convulsions and chronic insomnia [2]. BZD drugs increase the effects of a natural chemical called gamma-aminobutyric acid on the body and subsequently reduce the activity in the areas of the brain. BZDs have proven effective when used only once or as doses for a period of time not exceeding several weeks. BZDs usually contain a seven-membered ring connected to an aromatic ring, with four main substituent groups that can be changed, giving a change for the synthesis of several derivatives (Figure 1). Therefore, developing a dependable analytical method including BZDs separation and quantification needs to be considered.

| Benzodiazepine | R₁ | R₂ | R₃ | R₅ | R₇ |
|---|---|---|---|---|---|
| Bromazepam | H | = O | H | 2'-Pyridyl | Br |
| Clonazepam | H | = O | H | 2-Cl-Phenyl | NO₂ |
| Lorazepam | H | = O | OH | 2-Cl-Phenyl | Cl |
| Nordiazepam | H | = O | H | Phenyl | Cl |
| Diazepam | CH₃ | = O | OH | Phenyl | Cl |

**Figure 1.** Chemical structures of the five benzodiazepines (BZDs) studied drugs.

Several separation methods for the quantification of BZD drugs in pharmaceutical preparations were reported [3–8]. Regardless of the important therapeutic applications of BZDs, it is used in an inappropriate way in addition to some substances by drug addicts [3]. It has become very important in forensic research and toxicology institutions to know selective analytical methods for the determination of BZDs in the human plasma due to the complexity of the matrix and the diversity of BZDs in the market having different and wide toxicity (overdose) ranges [9–13]. The forensic labs usually received the blood samples, which were randomly withdrawn from humans, e.g., drivers on the road, and the response should be given within a few hours. Therefore, in the current work, we focused on this demand. We developed a simultaneous analytical method for the screening and quantification of some BZDs, which are commonly used in the market, within their therapeutic concentrations in the blood samples.

HPLC/DAD methods have gained a wide position today due to their simplicity and being inexpensive and reliable, making them the ideal choice for the quantitative analysis and separation of BZDs [11–14]. Referring to these previous methodological developments, some of them included unique high-cost columns (monolith [12] and semi-micro [13]), and the other used a C8 column [11]). Among the procedures followed for treating blood samples is the liquid–liquid extraction, including expensive chemicals or solid-phase extraction with tedious and time-consuming operations [12–14]. Moreover, these published methods determined BZDs by acidic mobile phases with lower pH values than 2.5 [12,13].

Therefore, in the current work, a simple RPLC/DAD method using the most common C18 column for the simultaneous determination of five BZDs (bromazepam, clonazepam, diazepam, lorazepam, and nordiazepam) was developed in a neutral mobile phase. The validated proposed RPLC/DAD method was successfully applied to the human plasma matrix after a simple liquid–liquid extraction by methanol, which is compatible with the mobile phase composition. This RPLC method was to minimize the complexity of BZDs analysis in the human plasma matrix while keeping the sensitivity and selectivity required for their detection. Therefore, the current developed method could be used for a simple and reliable simultaneous analysis of BZDs in any forensic lab for toxicological analysis.

## 2. Materials and Methods

### 2.1. Chemicals and Reagents

Bromazepam (96% purity), clonazepam (98% purity), diazepam (98% purity), lorazepam (99% purity), and nordiazepam (97% purity) standards were supplied by Poison Control and Forensic Medical Chemistry Center (Jeddah, Saudi Arabia). HPLC grades of acetonitrile (ACN) and methanol (MeOH) were obtained from Sigma-Aldrich (Steinheim, Germany). Sodium hydroxide was purchased from Qualikems Fine Chem Pvt. Ltd., (Nandesar, Vadodara, India). Potassium hydrogen phosphate and dipotassium hydrogen phosphate were obtained from BDH Chemicals Ltd., (Poole, London, UK). A Milli-Q water purification system from Millipore (Millipore, Bedford, MA, USA) was used to purify the water used in the research and bring it to high levels of purity by following the method of reverse osmosis. Human plasma samples were collected from the Blood Donation Unity of King Abdulaziz University Hospital (Jeddah, Saudi Arabia).

### 2.2. Instrumentation

All chromatographic measurements were made on an Agilent HPLC 1290 infinity system equipped with a diode-array detector (DAD). An agilent quaternary pump and automatic sampler were used. Automatic sample injection was performed via a Rheodyne 7725i injection valve (Rheodyne, Cotati, CA, USA) equipped with a 20 µL loop. Mettler Toledo analytical balance (Columbus, OH, USA), Hettich centrifuge (Model D78532, Tuttlingen, Germany), Agitator orbital Shaker (Ivy men, Spain), sonicator (Model LUC-405/410/420, Seoul, Korea), and Mettler Toledo pH meter (Columbus, OH, USA) were used for the preparation of standard and sample solutions.

### 2.3. Chromatographic Procedure

The separation was accomplished using a Symmetry C18 column with a 150 × 4.6 mm i.d. and a 5 µm particle size (WATER, Williamsburg, VA, USA). A phosphate buffer (pH 7.0, 20 mmol $L^{-1}$) and methanol (50:50, $v/v$) was used as the neutral isocratic mobile phase, which was filtered before use by a 0.45 µm membrane filter (Supelco, Bellefonte, PA, USA). The flow rate, wavelength, and column temperature were fixed at 1.0 mL $min^{-1}$, 214 nm and 40 °C, respectively.

### 2.4. Sample Preparation

#### 2.4.1. Standard Solutions

Standard stock solutions of bromazepam (Log $P_{o/w}$ = 2.05), clonazepam (Log $P_{o/w}$ = 2.41), diazepam (Log $P_{o/w}$ = 2.82), lorazepam (Log $P_{o/w}$ = 3.5) and nordiazepam (Log $P_{o/w}$ = 2.93) with concentrations of 1000 µg $mL^{-1}$ were prepared in methanol. The solutions of all compounds were thoroughly mixed and stored at 4 °C, which were stable for a period of time of more than one month. Working solutions of all drugs were prepared daily by dilution in methanol.

#### 2.4.2. Phosphate Buffer

The phosphate buffer was prepared by dissolving the amount of 1.330 g $KH_2PO_4$ and 2.778 g $K_2HPO_4$ all together in 1000 mL of distilled water, giving a pH of 7.0. Buffer was usually prepared every day and was immediately filtered before use.

#### 2.4.3. Human Plasma Samples

The treatment of real human plasma matrices was carried out by liquid–liquid extraction (methanol), which is still used as a simple and reliable sample treatment technique [15]. Donors were selected on condition with healthy volunteers aged between 30 and 45 years (*n* = 5). After a sample was taken from a donor, a system of identification and tracking was followed to ensure that the sample is correctly matched with the donor. Blood was collected in tubes containing EDTA and placed immediately on ice after collection. Immediately, three concentration levels of 70, 200, and 400 ng $mL^{-1}$ of standard solutions were prepared

in 200 µL of drugs-free plasma collected solution. Then, protein removal was performed by sedimentation with the addition of 1000 µL of methanol [16–18]. After vortexing for 30 s and centrifugation for 5 min at 5000 rpm, the clear supernatant was transported to glass tubes and then was dried under a nitrogen flow at 30 °C to 50 µL volume. After that, 100 µL of the mobile phase was added to the remained dried solution. A total of 20 µL was injected into the HPLC system.

## 3. Results and Discussion

### 3.1. Method Optimization

#### 3.1.1. Selection of Column

Several common C18 columns such as an SB-C18 ZORBAX column (250 mm × 4.6 mm × 5 µm), PRONTOSIL EUROBOND C18 (125 mm × 4.6 mm × 5 µm) and Symmetry C18 (150 mm × 4.6 mm × 5 µm) were studied. The best resolution (ranged from 1.6 to 4.5) among the separated BZDs was achieved by Symmetry C18 column (150 mm × 4.6 mm × 5 µm).

#### 3.1.2. Selection of Mobile Phase Composition

Several mobile phase combinations were tested by mixing different concentrations of phosphate buffer (10–50 mmol L$^{-1}$) with methanol or acetonitrile organic modifier at different ratios ranging from 30 to 55% (*v/v*) in the presence of 40 °C column temperature. It was found that the use of acetonitrile led to a loss of the resolution between clonazepam and bromazepam. However, increasing the methanol ratio from 30% to 50% reduced the total analysis time of the five compounds to 20 min while maintaining an acceptable resolution of more than 1.6 among the studied drugs. The best ratio was decided to be 50:50 (*v/v*) of 20 mmol L$^{-1}$ phosphate buffer and methanol, as shown in Figure 2.

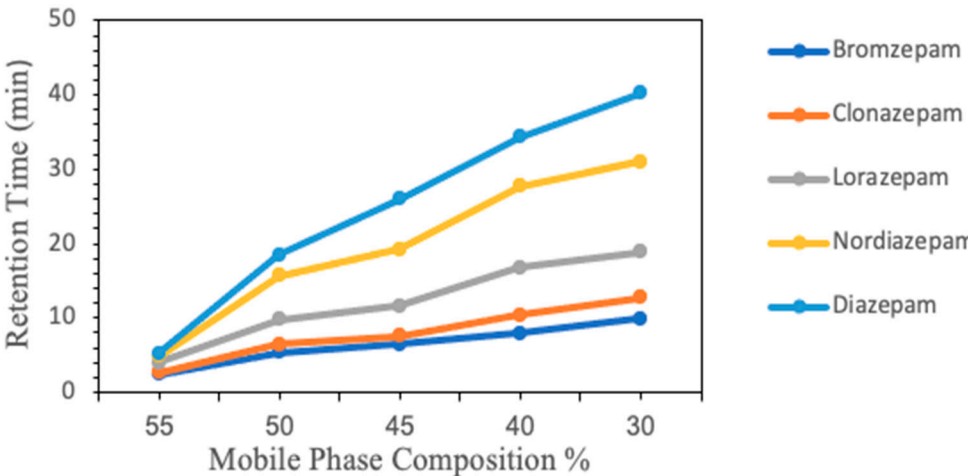

**Figure 2.** Effect of methanol composition in the mobile phase on the retention times of selected BZDs. Other experimental conditions as described in the text.

#### 3.1.3. Selection of Mobile Phase pH

Adjusting the mobile-phase pH can be a powerful tool for obtaining RPLC separations. Several pH values ranging from 5.0 to 9.0 were tested against the column efficiency (number of theoretical plates per meter

$$N = 5.545 \{ t_r \div w_{1/2} \}^2$$

where $t_r$ is the retention time and $w_{1/2}$ the half peak width), as shown in Figure 3. The effect of pH on retention and peak shape is related to the ionization character of solutes and the presence of residual silanols in the stationary phase (This is because, with silica columns based, a pH that is lower than two could lead the silica to become hydrolyzed, while a

pH that is higher than 8 could lead to the solubility of the silica [19]). The pKa values of selected BZDs were bromazepam (2.9 and 11.0), clonazepam (1.5 and 10.5), lorazepam (1.3 and 11.5), nordiazepam (3.5 and 12.0) and diazepam (3.1). It is clear that BZD drugs either have one pKa value or two pKa values. The first pKa value refers to the deprotonation of the nitrogen cation at position 4. The second pKa refers to the deprotonation of the nitrogen atom at position 1, as shown in Figure 1 [14]. The deprotonation of the nitrogen atom at position 1 is thought to be resonance stabilized with the negatively charged oxygen atom [14]. The presence of an electron-withdrawing ortho-chlorine substituent on the phenyl ring decreases the pKa1 value. Clonazepam and lorazepam have this ortho-chlorine substituent and have a calculated pKa1 value less than or equal to 1.5.

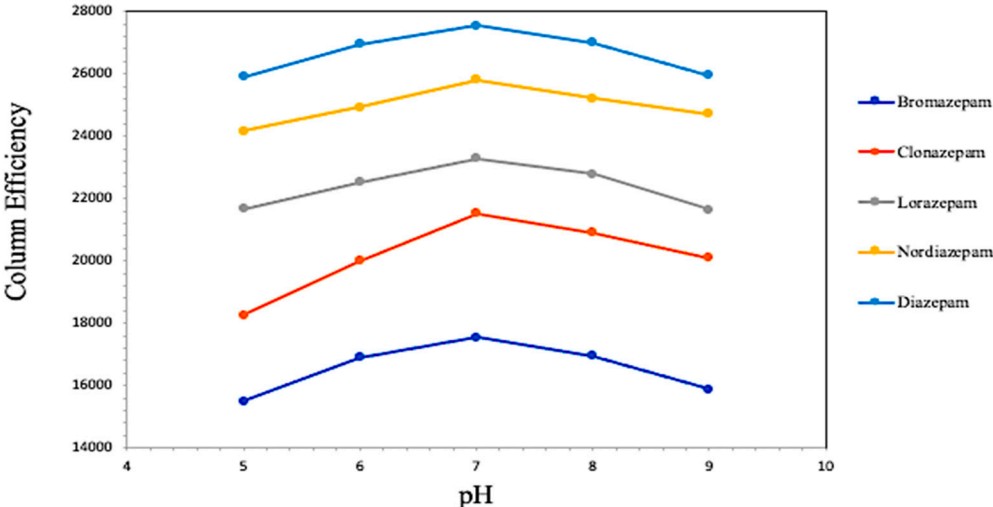

**Figure 3.** The change of column efficiency of selected BZDs at different pH values of the mobile phase. Other experimental conditions as described in the text.

The buffer pH range from 5.0 to 9.0 was chosen in this study in order to confirm the presence of full or a majority of non-ionized forms, which increased the drugs' affinity to the stationary phase and thus improved their peaks' separation and symmetry. Figure 3 shows that working at buffer pH 7.0 led to an increase in the column efficiency (ranged from 16,000 to 26,000 N/m) and asymmetry factor (AS) (bromazepam = 1.00, clonazepam = 1.05, diazepam = 1.20, lorazepam = 1.14, and nordiazepam = 1.00), which is probably due to its effect on the sharpness of the peaks. On the other hand, it is well known for the most robust retention conditions, the mobile phase pH should be beyond 1.5 units from the compound's pKa, where ionization is entirely suppressed [19,20]. Therefore, buffer pH 7.0 was chosen as the optimum pH for the simultaneous separation of selected BZD drugs.

3.1.4. Selection of Column Temperature, Flow Rate, and Wavelength

Several column temperature values were studied from 30 to 60 °C. A reduction in the total analysis time required for the complete separation of studied drugs with sharper peaks was observed by increasing the column temperature. However, after 40 °C, there was a significant decrease in the column efficiency and separation resolution between BZD drugs. Therefore, 40 °C was chosen as the optimal separation column temperature. Furthermore, studying the flow rate from 0.5 to 1.2 mL min$^{-1}$ showed a decrease in the column efficiency. However, 0.3 mL/min seems to be a good compromise when considering the resolution between peaks. At this flow rate, the retention time reproducibility (*n* = 6) was always better than 1%. Raising the column temperature to 40 °C at a flow rate of 1.0 mL min$^{-1}$ led to a decrease in the total analysis time of the studied compounds to 20 min, and the peak shapes were also improved (Figure 4). Moreover, the ultraviolet spectra of the studied BZDs showed the appropriate maximum absorption wavelength at 214 nm to achieve the highest sensitivity for the studied compounds simultaneously.

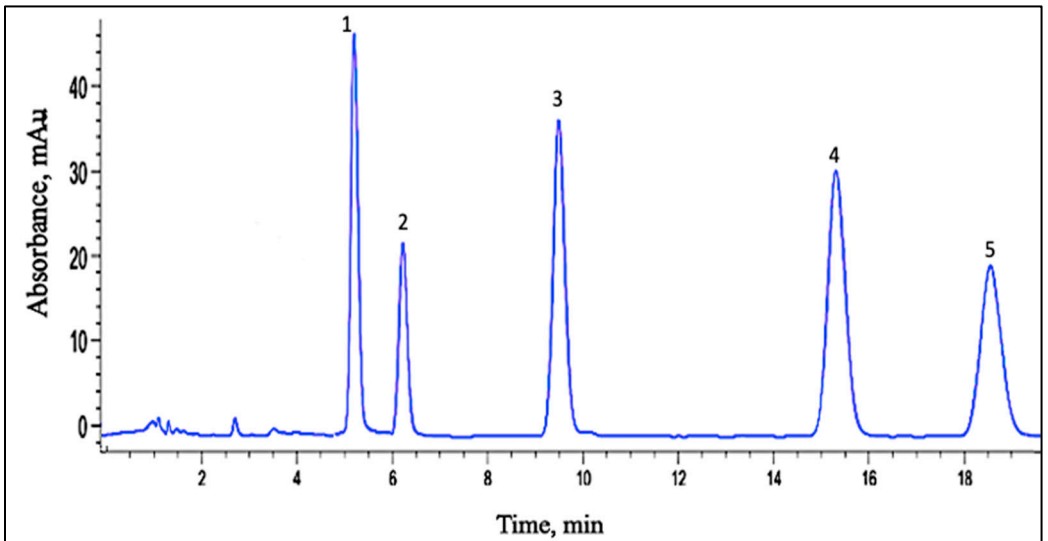

**Figure 4.** Chromatogram of BZDs mixture at the optimal conditions of mobile phase containing 50% of 20 mmol $L^{-1}$ phosphate buffer at pH = 7 and 50% methanol, flow rate at 1.0 mL $min^{-1}$, a wavelength at 214 nm and 40 °C column temperature. BZDs mixture: 1-Bromazepam (170 ng $mL^{-1}$), 2-Clonazepam (100 ng $mL^{-1}$), 3-Lorazepam (130 ng $mL^{-1}$), 4-Nordaizepam (160 ng $mL^{-1}$) and 5-Diazepam (130 ng $mL^{-1}$).

Figure 4 shows the obtained chromatogram with a baseline separation among the selected BZDs after the application of optimum conditions. The described RPLC/DAD method was established using simple experimental conditions to present a quick quality control procedure for the quantification of BZDs.

*3.2. Method Validation*

The current developed RPLC method was validated based on the International Conference on Harmonization guidelines (ICH) in terms of selectivity, linearity, precision, accuracy, sensitivity, and robustness [21].

3.2.1. Selectivity

Selectivity is defined as the capability of the current method to detect the analyte in the existence of potential impurities and other similarities. The peak identity of each drug in the proposed method was confirmed by running the individual analyte as shown in the Supporting Information (S1 to S5). Moreover, each analyte was determined by spiking five different concentrations of each drug standard solution in the studied plasma matrix giving linearity equations similar to calibration equations at the same retention times. Furthermore, photodiode-array detection offered valuable assets for peak identification and peak purity determination. The obtained results proved that BDZs were free of interferences and the absence of any peak in the blank plasma related to any of the studied analytes.

3.2.2. Linearity

Linearity was measured by establishing the calibration curves [21]. Here, 25 mL volumetric flasks were used to prepare stock solutions of BZDs in methanol to produce 10 mixture solutions for each drug. The curves of selected BZDs were constructed using peak area versus concentrations of the analytes, which were measured in triplicate for each concentration. For a plasma matrix, the same working solutions were used. The least-square linear regression equations were given in Table 1 together with the corresponding $R^2$ values. The calibration curves for standard and plasma samples were found to be linear in the concentration range of 5–500 ng $mL^{-1}$ for bromazepam and diazepam, 3–500 ng $mL^{-1}$ for clonazepam and lorazepam and 1–500 ng $mL^{-1}$ for nordiazepam. The coefficients of

determination ($R^2$) were greater than 0.9992 and 0.9970 for all standard and plasma sample compounds, respectively.

**Table 1.** Linearity and sensitivity data for the proposed method (x = ng mL$^{-1}$, *n* = 3) in standard and real samples.

| Analytes | Linearity Equation | $R^2$ | LLOQ (ng mL$^{-1}$) | LOD (ng mL$^{-1}$) |
|---|---|---|---|---|
| | | Standard samples | | |
| Bromazepam | y = 5.25x − 30.50 | 0.9992 | 5.09 | 1.55 |
| Clonazepam | y = 13.00x − 51.60 | 0.9993 | 4.38 | 2.32 |
| Lorazepam | y = 14.56x − 90.80 | 0.9996 | 3.81 | 1.26 |
| Nordiazepam | y = 18.59x − 81.90 | 0.9999 | 1.78 | 0.54 |
| Diazepam | y = 13.68x − 58.20 | 0.9998 | 5.59 | 1.79 |
| | | Plasma real samples | | |
| Bromazepam | y = 5.57x + 27.34 | 0.9970 | 6.34 | 2.09 |
| Clonazepam | y = 6.86x + 10.55 | 0.9990 | 7.65 | 1.44 |
| Lorazepam | y = 4.39x + 28.53 | 0.9990 | 4.16 | 1.26 |
| Nordiazepam | y = 7.28x + 11.85 | 0.9980 | 4.77 | 1.57 |
| Diazepam | y = 6.32x + 30.022 | 0.9970 | 5.92 | 1.85 |

### 3.2.3. Extraction Recovery

The recovery of the liquid–liquid extraction procedure was assessed by analyzing extracts of spiked plasma samples containing the studied BZDs compounds. Aliquots of 20 mL of plasma solutions at the concentration levels 46.7, 133.3, and 266.7 ng mL$^{-1}$ of BZDs were injected into the HPLC system, and triplicate measurements (*n* = 3) were recorded for each concentration. The extraction recovery values of selected BZDs were calculated by comparing the analyte peak areas obtained from plasma samples with those in standard methanolic solutions. The results were expressed as percent recoveries. All values were found within the acceptable recovery range from 96.5 to 107.5%.

### 3.2.4. Sensitivity

The sensitivity of the method was tested by examining the limit of detection (LOD) values that were calculated based on the standard deviation of the intercept and the slope of the calibration curve (LOD = 3.3 σ/S, where σ is the standard deviation (SD) of the intercept and S is the slope of the calibration curve). The LOD values for selected BZDs were 1.55 and 2.09 ng mL$^{-1}$ (Bromazepam), 2.32 and 1.44 ng mL$^{-1}$ (Clonazepam), 1.26 and 1.26 ng mL$^{-1}$ (Lorazepam), 0.54 and 1.57 ng mL$^{-1}$ (Nordiazepam) and 1.79 and 1.85 ng mL$^{-1}$ (Diazepam), respectively, for standard and plasma samples. Furthermore, the lower limit of quantification (LLOQ) is the lowest concentration at which the target analytes can be reliably measured and reported with a certain degree of confidence [21]. The LLOQ values were calculated and found to be 5.09 and 6.34 ng mL$^{-1}$ (Bromazepam), 4.38 and 7.65 ng mL$^{-1}$ (Clonazepam), 3.81 and 4.16 ng mL$^{-1}$ (Lorazepam), 1.78 and 4.77 ng mL$^{-1}$ (Nordiazepam) and 5.59 and 5.92 ng mL$^{-1}$ (Diazepam), for standard and plasma samples, respectively. These results are summarized in Table 1. Nevertheless, values of detection and quantification limits for the five BZDs compounds indicated a good sensitivity of the method.

### 3.2.5. Precision

The precision of this RPLC method in plasma was estimated by calculating the intraday precision (repeatability), which has been measured in triplicate over the same day at three concentration levels (50, 200, and 400 ng mL$^{-1}$), and its relative standard deviations (RSDs) found were lower than 1%, as shown in Table 2. The interday precision (reproducibility) was measured with repeated analysis (*n* = 9) at the same concentrations over a period of three days. The interday RSDs calculated were lower than 4% (Table 2).

**Table 2.** Accuracy expressed as recovery (%) and precision, expressed as relative standard deviation (RSD, %), of the developed RPLC methods for the analysis of selected BZDs.

| Analytes | Amount Added (ng mL$^{-1}$) | Amount Measured (ng mL$^{-1}$) | Intraday Accuracy (%, $n = 3$) | Interday Accuracy (%, $n = 9$) | Intraday Precision (%, $n = 3$) | Interday Precision (%, $n = 9$) |
|---|---|---|---|---|---|---|
| Bromazepam | 50 | 47.6 | 95.2 | 90.5 | 0.320 | 1.012 |
| | 200 | 203.1 | 101.6 | 101.9 | 0.285 | 2.112 |
| | 400 | 398.9 | 99.7 | 100.7 | 0.124 | 3.321 |
| Clonazepam | 50 | 50.1 | 100.2 | 100.8 | 0.222 | 0.018 |
| | 200 | 200.3 | 100.1 | 106.1 | 0.147 | 1.168 |
| | 400 | 399.5 | 99.9 | 100.9 | 0.155 | 3.278 |
| Lorazepam | 50 | 51.7 | 103.4 | 105.3 | 0.078 | 1.043 |
| | 200 | 197.8 | 98.9 | 103.4 | 0.045 | 2.164 |
| | 400 | 400.5 | 100.1 | 102.5 | 0.036 | 3.672 |
| Nordiazepam | 50 | 51.6 | 103.2 | 108.8 | 0.276 | 0.072 |
| | 200 | 197.4 | 98.7 | 104.2 | 0.143 | 1.256 |
| | 400 | 400.7 | 100.2 | 105.5 | 0.024 | 2.432 |
| Diazepam | 50 | 50.0 | 100.0 | 105.3 | 0.181 | 0.088 |
| | 200 | 200.1 | 100.1 | 100.1 | 0.154 | 1.568 |
| | 400 | 400.5 | 100.1 | 100.1 | 0.046 | 2.884 |

### 3.2.6. Accuracy

The accuracy of the RPLC method was evaluated by performing analyses of selected BDZs samples at three concentration levels (50, 200, and 400 ng mL$^{-1}$, intraday accuracy $n = 3$) in plasma samples versus a calibration curve in standard solutions. The results found from the accuracy study indicated high recoveries of BDZs by the proposed method: 95.2–103.4%, as shown in Table 2. Moreover, the interday accuracy ($n = 9$) was evaluated by repeating the same three concentrations within three consecutive days giving acceptable recoveries in the range of 90.5–108.8%.

### 3.2.7. Stability of BZDs in Plasma Samples

The stability of sample solutions was examined by the proposed RPLC method over a period of 30 days. The freshly prepared solutions at room temperature, and the 30 days stored samples in a refrigerator, were analyzed. Biological plasma samples spiked with 50 ng mL$^{-1}$ of each BZDs were subjected to deproteinization and stored at ambient temperature and in a freezer at 4 °C for 30 days. Short time stability was estimated after 12 h at room temperature, 24 h of storage in a refrigerator, and for long-term assay after 5, 7, and 30 days refrigerated. Each sample was analyzed for intact BZDs compounds once daily after a freeze–thaw cycle to investigate stability. Comparing the recovery and RSD results between the samples currently prepared and the samples measured after 30 days proved that those kept at a temperature of 4 °C are stable and can be used without being afraid of any decomposition products or interferences as cited (recovery, %) in the Supporting Information (Table S1). This proved that the plasma samples could be used for 30 days without deterioration.

### 3.2.8. Robustness

Robustness is the measure of the ability of an analytical method to stay unchanged by small but intentional variations in experimental parameters [22]. The flow rate and column temperature were the parameters varied by small changes as the most effective means to evaluate the current method's robustness. The selected variable parameters were column temperature (39.5 °C, 40 °C and 40.5 °C) and flow rate (0.95 mL min$^{-1}$, 1.00 mL min$^{-1}$ and 1.05 mL min$^{-1}$), as shown in Table 3. The retention factor, resolution, selectivity factor, and column efficiency (number of theoretical plates per meter) were evaluated with these

small changes. Furthermore, the effect of a small change in pH ($\pm$0.05) and mobile phase compositions ($\pm$0.5%) around their optimal values were also evaluated. It was found that simple changes made to the method did not produce any noticeable changes, and the results were excellent in most conditions. These results are expected due to the robustness of retention of non-ionized species of analytes at a neutral pH value by RPLC. Variation in the experimental parameters and carrying out the experiment at room temperature indicated its reliability during normal use and concluded that the proposed method was robust.

**Table 3.** Robustness of the selected BZDs under various conditions of flow rate and column temperature.

| Chromatographic Parameter | Analyte | Nominal Condition * | Flow Rate, mL min$^{-1}$ | | Column Temperature, °C | |
|---|---|---|---|---|---|---|
| | | | 0.95 | 1.05 | 39.5 | 40.5 |
| Retention Time ($t_R$) | Bromazepam | 5.2 | 5.6 | 5.0 | 5.2 | 5.2 |
| | Clonazepam | 6.3 | 6.7 | 6.0 | 6.3 | 6.3 |
| | Lorazepam | 9.5 | 10.1 | 9.1 | 9.6 | 9.5 |
| | Nordiazepam | 15.2 | 16.1 | 14.7 | 15.4 | 15.2 |
| | Diazepam | 18.4 | 19.4 | 17.8 | 18.7 | 18.4 |
| Theoretical Plates (N/m) | Bromazepam | 16,842 | 19,868 | 17,562 | 14,248 | 16,118 |
| | Clonazepam | 20,749 | 23,464 | 21,249 | 29,593 | 19,544 |
| | Lorazepam | 22,426 | 28,602 | 24,453 | 19,950 | 22,690 |
| | Nordiazepam | 25,151 | 31,813 | 28,851 | 24,089 | 25,054 |
| | Diazepam | 26,368 | 32,131 | 29,085 | 23,854 | 28,123 |
| Retention Factor ($k$) | Bromazepam | 3.7 | 3.7 | 3.7 | 3.7 | 3.7 |
| | Clonazepam | 4.6 | 4.7 | 4.7 | 4.7 | 4.7 |
| | Lorazepam | 7.5 | 7.5 | 7.6 | 7.6 | 7.5 |
| | Nordiazepam | 12.6 | 12.7 | 12.8 | 12.8 | 12.7 |
| | Diazepam | 15.5 | 15.5 | 15.7 | 15.8 | 15.6 |
| Resolution ($R_S$) | Bromazepam | 1.6 | 1.7 | 1.6 | 1.7 | 1.5 |
| | Clonazepam | 3.7 | 4.1 | 3.8 | 3.9 | 3.7 |
| | Lorazepam | 4.5 | 5.0 | 4.8 | 4.3 | 4.5 |
| | Nordiazepam | 1.9 | 2.0 | 2.0 | 1.8 | 1.9 |
| | Diazepam | - | - | - | - | - |
| Selectivity factor ($\alpha$) | Bromazepam | 1.3 | 1.3 | 1.3 | 1.3 | 1.3 |
| | Clonazepam | 1.6 | 1.6 | 1.6 | 1.6 | 1.6 |
| | Lorazepam | 1.7 | 1.7 | 1.7 | 1.7 | 1.7 |
| | Nordiazepam | 1.2 | 1.2 | 1.2 | 1.2 | 1.2 |
| | Diazepam | - | - | - | - | - |

* Nominal conditions: flow rate 1.00 mL min$^{-1}$ and column temperature 40.0 °C. Number of theoretical plates per meter (N/m) = 5.545 $(t_R/W_{50\%})^2$. Retention factor ($k$) = $(t_R - t_0)/t_0$. Resolution of two given peaks (Rs): Rs = 2 $(t_{R2} - t_{R1})/(W_1 + W_2)$. Selectivity factor ($\alpha$) = $k_2/k_1$ = $(t_{R2} - t_0)/(t_{R1} - t_0)$.

### 3.2.9. Comparison with the Previous Published Methods

The current work aimed to determine the subject BZDs in a plasma matrix. Therefore, the current comparison is concerned with the previous publications of BZD in a plasma matrix by HPLC, as presented in Table 4. In order to overcome the difficult and multiple steps extraction procedures, this work developed a simple and reliable liquid–liquid extraction. Some of the published methods required the use of an ultrasonic bath, which is considered one of the noise pollution sources in labs. Others used the cartridges for extraction, which need to be conditioned several times before use with the organic solvent, and again, it was washed several times after the analytes passing [23–34]. However, the proposed RP-HPLC method is less technically demanding and easily completed with a few microliters of methanol. Moreover, the current method achieved LOD values in plasma matrix between 1.26 and 2.09 ng mL$^{-1}$ that overcome most of the previous methods, as shown in Table 4. Furthermore, the determination of BZDs in the presence of their metabolites by HPLC/DAD on a C18 column was validated elsewhere [32,34,35]. Diazepam and its metabolites N-desmethyldiazepam were separated with high resolution in human plasma [32]. Clonazepam and its metabolites 7-aminoclonazepam were well separated: clonazepam at 34.380 min and 7-aminoclonazepam at 12.924 [34]. Bromazepam and its common metabolites 2-(2-amino-5-bromobenzoyl) pyridine were easily separated at retention times of 4.397 and 5.317 min, respectively [35]. Nordiazepam, which is classified as diazepam metabolites from the effect of liver enzymatic behavior, as mentioned in [32], was

separated in the current method with a good resolution of 1.9. Furthermore, a surfactant was used as an additive in the mobile phase without requiring sample treatment [9,35,36]; however, the current method has also used a simple and reliable liquid–liquid methodology for the sample treatment [15].

**Table 4.** Comparison between the current method and the previously published methods.

| Drug | Extraction Procedure | Column Used | LOD ng/mL | Ref. |
|---|---|---|---|---|
| -Alprazolam -Bromazepam -Brotizolam -Chlordiazepoxide-Clobazam -Clonazepam -Clotiazepam -Delorazepam -Diazepam -Flunitrazepam -Flurazepam -Lorazepam -Lormetazepam -Oxazepam -Triazolam | The solid phase extraction (SPE) procedure was carried out on cartridges that had been washed several times by methanol before the sample passing for conditioning. | Waters (Milford, USA (XTerra C8 RP column (150 × 4.6 mm id, 5 μm) | 1.00–2.50 | [23] |
| -Chlordiazepoxide -Diazepam | Expensive chemicals such as zinc acetate nanoparticles and thioacetamide were used. The analytes went through several steps including shaken with a vortex and using an ultrasonic bath for a long time. It was transferred to another glass pipette tip whose narrow side is capped by a three-layer handmade filter. | Zorbax SB-C18 (150 mm × 4.6 mm, 5 μm) (Agilent) column | 1.2 Chlordiazepoxide 1.5 Diazepam | [24] |
| -Alprazolam -Bromazepam -Clonazepam -Diazepam -Lorazepam -Lormetazepam -Tetrazepam | Dispersive liquid–liquid microextraction (DLLME) needs two organic solvents, one as a disperser (methanol) and the another as an extraction solvent (chloroform). The pH of the media should be controlled at pH 9; shaking for a few minutes is required using an ultrasonic water bath. The sedimented phase was retrieved using a syringe and redissolved in the mobile phase. | Shield RP18 and C18 columns | 1.7–5.3 | [25] |
| -Alprazolam -Bromazepam -Diazepam -Flunitrazepam | A DSC-18 (Supelco) cartridge preconditioned with methanol was used. After sample application, benzodiazepines were eluted with methanol. Dry residues were reconstituted with 200 μL of methanol. ACN (200 μL) was added to aliquots of 50 μL of pooled blood plasma containing 200 μL of methanol for blank samples. | C8 analytical (250 mm 64 mm, 5 μm) column | 3.3–10.2 | [26] |
| -Flunitrazepam -Clonazepam -Oxazepam -Lorazepam -Chlordiazepoxide -Nordiazepam -Diazepam | Solid-phase extraction using a C18 cartridge, which was activated with methanol. | C18 DB column (250 mm × 4.6 mm, 5 μm) | 20 for all | [27] |

**Table 4.** *Cont.*

| Drug | Extraction Procedure | Column Used | LOD ng/mL | Ref. |
|------|----------------------|-------------|-----------|------|
| -Bromazepam<br>-Clonazepam<br>-Diazepam<br>-Flunitrazepam<br>-Lorazepam<br>-Alprazolam<br>-A-hydroxyalprazolam<br>-A-hydroxytriazolam | Conditioned C18 cartridge with methanol was used. After washing with water, retained drugs were eluted with (261) mL MeOH/CH3 CN (50:50). The residue was reconstituted with methanol prior to their injection into the LC system. | C8 (250 mm 65 mm, 5 μm) analytical column | 0.02–0.47 | [28] |
| -Alprazolam<br>-Bromazepam<br>-Diazepam<br>-Lorazepam | Fabric phase sorptive extraction medium was conditioned by acetonitrile:methanol (50:50 *v/v*) for 2 min. The FPSE was handled using tweezers (to avoid touching).<br>Then, it was inserted into a vial with 500 μL CH3CN:CH3OH (50:50 *v/v*) for 10 min. The eluate was evaporated and reconstituted using 50 μL CH3CN:CH3OH (50:50 *v/v*). | LiChrospher®100 RP-C18 (5 μm, 250 × 4m) analytical column | 0.01 for all | [29] |
| -Alprazolam<br>-Clonazepam<br>-Diazepam | Ammonium formate was mixed with the serum (pH = 8.6) and 3 mL of diethylether. After vortexing and centrifuging, the supernant layer was transferred and evaporated to dryness. The residue was dissolved in mobile phase in an ultrasonic bath and filtered before injection into the HPLC system. | RP-18 column (70 mm × 4.6 mm and 5 mm) | 8–27 | [30] |
| -Alprazolam<br>-Bromazepam<br>-Diazepam<br>-Lorazepam<br>-Lormetazepam<br>-Tetrazepam | The microwave-assisted extraction conditions were optimized for the extraction of BZDs from human plasma. The samples were extracted at 89 °C for 13 min, using 8 mL of chloroform/2-propanol (4:1, *v/v*). The extracts were redissolved in the mobile phase and then injected into HPLC. | RP8 (250 mm × 4.6 mm inner diameter, 5 μm particle size | 6.2–12.6 | [31] |
| Diazepam | The treatment of plasma samples was carried out by liquid–liquid extraction (LLE) using an Eppendorf polypropylene tube and then extracted with toluene. After vertical agitation (5 min) and centrifugation (10,000 rpm, 2 min), the upper organic layer was evaporated under a gentle stream of air. The dried extract was reconstituted in the mobile phase and 100 μL of aliquot was injected into HPLC. | RP18 (100 mm × 4.6 mm column) | 2 | [32] |
| Nitrazepam<br>Midazolam | A mixture of SDS, Tween 80 as an emulsifier, and 1-undecanol as extraction solvent were added to the sample solution. Then, it was put in an ultrasonic water bath for 20.0 min, centrifuged and transferred into an ice bath for about 5.0 min. | A C18 column (4.5 mm × 150 mm with 5 μm particle size) | 0.017 Nitrazepam<br>0.086 Midazolam | [33] |
| -15 benzodiazepines, and some metabolites:<br>-7-aminonitrazepam,<br>-7-aminoflunitrazepam,<br>-7-aminoclonazepam | Diethylether and phosphoric acid were used for the liquid–liquid extraction, which took more than 30 min. | Nova-Pak phenyl column (5 prn particle size, 150 mm × 4.6 mm i.d.) protected by a Nova- Pak phenyl guard column (Millipore) | 10–100 | [34] |
| -Bromazepam<br>-Clonazepam<br>-Lorazepam<br>-Nordiazepam<br>-Diazepam | A simple liquid–liquid extraction was carried out as described in the experimental section. | Symmetry C18 column with a 150 × 4.6 mm i.d., 5 μm particle size (WATER, USA) | 2.09<br>1.44<br>1.26<br>1.57<br>1.85 | Current work |

### 3.2.10. System Suitability and Application to Human Plasma Matrix

A repeated injection of concentration of 50 ng mL$^{-1}$, including intra-inter day assessments for standard methanolic and plasma matrices, was used to measure the system's suitability. The precision with RSD values of peak areas 0.452–3.212% and that of retention times 0.026–0.077% were measured and indicated excellent suitability of the proposed method for application to the human plasma matrix. Three concentration levels of 70, 200, and 400 ng mL$^{-1}$ of standard analytes solutions were spiked to aliquots of blank human plasma solution. Table 2 shows the recovery results of the plasma matrix for each concentration after comparing their areas of the peaks with areas of standard solutions peaks, giving values within 95.2–103.4% (intraday) and 90.5–108.8% (interday). Figure 5 shows the chromatogram of a plasma matrix containing a mixture of 70 ng mL$^{-1}$ of bromazepam, clonazepam, lorazepam, nordiazepam, and diazepam in comparison with a blank plasma sample at the optimal conditions. This method has proven its reliability in analyzing several BZDs, including the most commonly used compounds in the market, without any interference with other compounds in the plasma matrix. Furthermore, the measurements of all BZDs therapeutic concentrations in plasma are in their ranges of linearity (1–500 ng mL$^{-1}$). The therapeutic ranges in blood of bromazepam, diazepam, clonazepam, lorazepam and nordiazepam are 80–170, 90–350, 100–400, 20–250 and 10–470 ng mL$^{-1}$ [21].

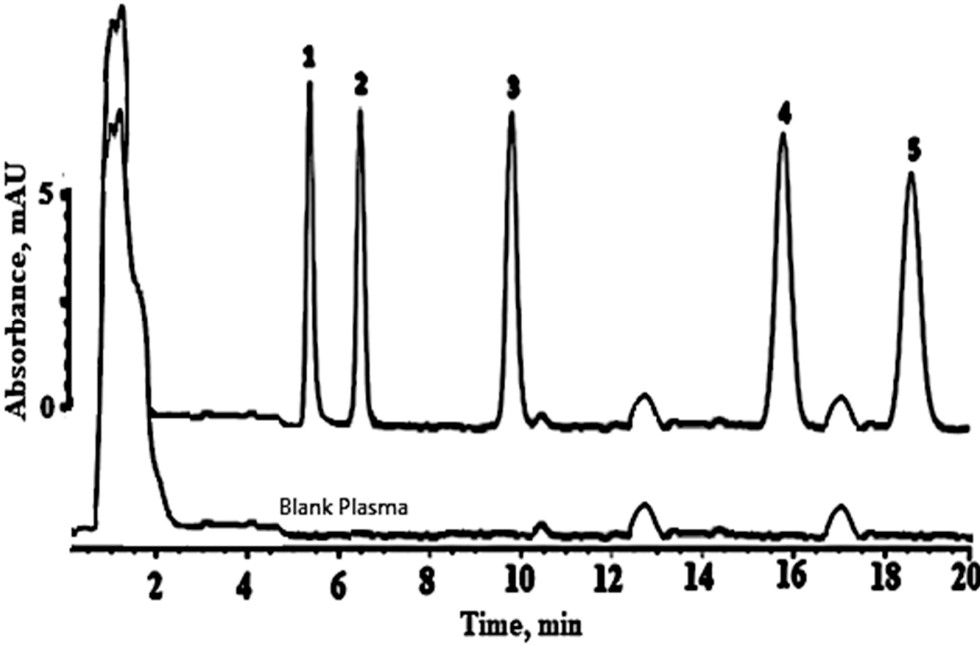

**Figure 5.** Chromatogram of a plasma matrix containing a mixture of 70 ng mL$^{-1}$ of bromazepam, clonazepam, lorazepam, nordiazepam, and diazepam compared with a blank plasma sample at the optimum conditions as described in Figure 4.

### 4. Conclusions

The current RPLC method was developed for the simple simultaneous determination of BZDs (bromazepam, clonazepam, lorazepam, nordiazepam, and diazepam) within their therapeutic concentration range in plasma matrix in order to be suitable for the reliable screening of these drugs in routine analysis. The selective extraction of analytes from the matrix was achieved by means of liquid–liquid extraction. Furthermore, photodiode-array detection offered valuable assets for peak identification and peak purity determination. The calibration curves were found to be linear in the concentration ranges of 5–500 ng mL$^{-1}$ for bromazepam and diazepam, 3–500 ng mL$^{-1}$ for clonazepam and lorazepam and 1–500 ng mL$^{-1}$ for nordiazepam. The LOD and LLOQ for selected BZDs ranged from 1.78 to 7.65 ng mL$^{-1}$ and from 3.81 to 6.34 ng mL$^{-1}$, respectively. The precision with RSD values of peak areas 0.452–3.212% and that of retention times 0.026–0.077%

indicated the suitability of the proposed method for application to plasma samples. The recovery percentages of real human plasma samples spiked with BZDs ranged between 95.2 and 103.4%. Moreover, the variation in the experimental parameters indicated its reliability during regular use, and the proposed method was robust. The current method could be appropriate for biological lab BZD drug quality control. It could also be a useful alternative simple analytical method in forensic toxicology. More future work will focus on modifying the current work with greener mobile phases, including the quantification of a larger window of BZDs compounds.

**Supplementary Materials:** The following supporting information can be downloaded at: https://www.mdpi.com/article/10.3390/analytica3020018/s1, Figure S1: Chromatogram of bromazepam, Figure S2: Chromatogram of clonazepam, Figure S3: Chromatogram of lorazepam, Figure S4: Chromatogram of nordiazepam, Figure S5: Chromatogram of diazepam, Table S1: Short and long term storage recoveries (%) and RSD.

**Author Contributions:** Funding acquisition, H.M.A.; Methodology, H.M.A. and D.A.E.-H.; Writing-review & editing, H.M.A.; data curation, N.A.A. and D.A.E.-H.; formal analysis, N.A.A.; funding acquisition, N.A.A.; investigation, N.A.A. and D.A.E.-H.; validation, N.A.A.; writing—original draft, N.A.A. and D.A.E.-H.; project administration, D.A.E.-H.; supervision, D.A.E.-H. All authors have read and agreed to the published version of the manuscript.

**Funding:** This research received no external funding.

**Institutional Review Board Statement:** Not applicable.

**Informed Consent Statement:** In this article the Human plasma samples were collected and handled using the ethics protocol reviewed and approved by the bioethical committee of King Abdulaziz University, under grant no. (G-248-166-1440).

**Data Availability Statement:** All data is contained within the article.

**Acknowledgments:** The authors would like to thank King Abdulaziz University for providing the appropriate laboratory for this research. The authors would also like to thank Poison Control and Forensic Medical Chemistry Center (Jeddah, Saudi Arabia) for providing the research group with the drugs stock solution and the Blood Donation Unity of King Abdulaziz University Hospital (Jeddah, Saudi Arabia) for the plasma sample suppling.

**Conflicts of Interest:** The authors declare no conflict of interest.

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
