# Peer review of "A Simple and Reliable Liquid Chromatographic Method for Simultaneous Determination of Five Benzodiazepine Drugs in Human Plasma"

_analytica, doi:10.3390/analytica3020018_

Round 1

Reviewer 1 Report

The presented MS deals with simultaneous determination of five benzodiazepine drugs in human plasma. The steps of the procedure are correctly assessed but I have the following comments to the presented results:

11. Please improve the Figures’ quality.

22. Please cite the references (procedures) used for the assessment of Linearity and LLOQ.

33. Inclusion of LODs of the current study in Table 4 will benefit the readers.

Author Response

First Review

The presented MS deals with simultaneous determination of five benzodiazepine drugs in human plasma. The steps of the procedure are correctly assessed but I have the following comments to the presented results:

1. Please improve the Figures’ quality.

Response: Thank you so much. We appreciate for your hard work earnestly, and we hope that the current revised form will meet with your approval. The figures quality have been improved and we hope it becom clear to the readers.

2. Please cite the references (procedures) used for the assessment of Linearity and LLOQ.

Response: Thank you so much. The references of LLOQ and linearity have been added in the revised text based on your recommendation.

3. Inclusion of LODs of the current study in Table 4 will benefit the readers.

Response: Thank you so much. The LODS of the current study have been cited in table 4 based on your recommendation.

Reviewer 2 Report

A simple and reliable liquid chromatographic method for simultaneous determination of five benzodiazepine drugs in human plasma” written by Hassan et al. established a method which can simultaneously determine five different drug molecules by a chromatographic method. In this article authors have performed many experiments to establish the method. Manuscript was written in a systematic way which is very understandable by the reader. Therefore I like to request editor to accept the manuscript for publication as it does perfectly go with the aims and objectives of this journal. My few minor corrections are below.

1.     Authors mentioned they have collected blood samples from five different healthy people, what is the basis they choose the donor, what is the procedure for blood collection, storage etc. have they performed they L-L extraction right after the blood collection? What solvent they have used for L-L extraction? Highly appreciate if they provide that information.

2.     In line, 116-118, are they add KH2PO4 and 2.778 g K2HPO4 into 1000-mL distilled water? Or all together gives 1000 mL?

3.     In line, 124 they need to provide a citation to verify their input that methanol initiate precipitation.

4.     In line, 284, and 311 please check the unit symbol.

5.     The quality of Figure 4, and 5 better to improve. Figure looks unclear. May be you can go with color to increase the visibility.

Author Response

“A simple and reliable liquid chromatographic method for simultaneous determination of five benzodiazepine drugs in human plasma” written by Hassan et al. established a method which can simultaneously determine five different drug molecules by a chromatographic method. In this article authors have performed many experiments to establish the method. Manuscript was written in a systematic way which is very understandable by the reader. Therefore I like to request editor to accept the manuscript for publication as it does perfectly go with the aims and objectives of this journal. My few minor corrections are below.

Response: Thank you so much. We appreciate for your hard work earnestly, and we hope that the current revised form will meet with your approval.

  1. Authors mentioned they have collected blood samples from five different healthy people, what is the basis they choose the donor, what is the procedure for blood collection, storage etc. have they performed they L-L extraction right after the blood collection? What solvent they have used for L-L extraction? Highly appreciate if they provide that information.

    Response: Thanks so much. All required information about donors, blood collection and storage, etc have been added in the revised text.

  2. In line, 116-118, are they add KH2PO4 and 2.778 g K2HPO4 into 1000-mL distilled water? Or all together gives 1000 mL?

    Response: Thank you so much. The phosphate buffer was prepared by dissolving the amount of 1.330 g KH2PO4 and 2.778 g K2HPO4 all together into 1000-mL distilled water. This information has been modified in the revised text

3. In line, 124 they need to provide a citation to verify their input that methanol initiate precipitation.
Response: Thank you so much. The required References based on your recommendation have been provided.

4. In line, 284, and 311 please check the unit symbol.

Response: Thank you so much. The unit symbol been corrected.
5. The quality of Figure 4, and 5 better to improve. Figure looks unclear. May be

you can go with color to increase the visibility.

Response: Thank you so much. We appreciate for your hard work earnestly. The quality of Figures 4 and 5 have been improved.

Reviewer 3 Report

In this study, a simple, sensitive and reliable RPLC method is developed for the determination of 5 benzodiazepinesin human plasma samples. A LLE procedure is used for the extraction of the drugs from the biological samples. The method ewas optimized and validated obtaining satisfactory results.

This developed method is a useful application, which could be used routinely, there are not particular novelties but the results achieved in terms of sensitivity, accuracy and reproducibility make it favourable to be used in forensic and clinical laboratories.

The manuscript is well organized AND the results clearly described.

Some minor comments:

1)      Add the titles of the articles cited in the references.

2)      Section 3.2.4, Table 3, specify that the results are referred to the standards and to the real samples, respectively.

3)      Section 3.2.7, in the supplementary material you can add a table with the recovery of the compounds in function of the short and long term storage.

Author Response

Comments and Suggestions for Authors

In this study, a simple, sensitive and reliable RPLC method is developed for the determination of 5 benzodiazepinesin human plasma samples. A LLE procedure is used for the extraction of the drugs from the biological samples. The method was optimized and validated obtaining satisfactory results. This developed method is a useful application, which could be used routinely, there are not particular novelties but the results achieved in terms of sensitivity, accuracy and reproducibility make it favourable to be used in forensic and clinical laboratories.

The manuscript is well organized AND the results clearly described.

1)

Response: Thank you so much. We appreciate for your hard work earnestly, and we hope that the current revised form will meet with your approval.

Some minor comments:

Add the titles of the articles cited in the references.

Response: Thanks so much. All titles of the articles have been cited in the references

2) Section 3.2.4, Table 3, specify that the results are referred to the standards and to the real samples, respectively.

Response: Thank you so much. It has been corrected

3) Section 3.2.7, in the supplementary material you can add a table with the recovery of the compounds in function of the short and long term storage.

Response: Thank you so much for your fruitful suggestions. Table. 1S in the supporting information file has been added to represent the recovery of the compounds in the short and long term storage.

This manuscript is a resubmission of an earlier submission. The following is a list of the peer review reports and author responses from that submission.

Round 1

Reviewer 1 Report

In this manuscript, the authors developed the HPLC-DAD method for the simultaneous determination of five benzodiazepines in human plasma. Different chromatographic conditions were also tested during method optimization.  

My remarks and recommendations are as follows:

  1. There are many methods of determining benzodiazepine drugs. The authors mentioned only some of them in the introduction. The presented results do not seem to be significantly new. The authors should clearly tell why their method is superior to the methods in the literature and should compare their results to the literature.
  2. Not only the column efficiency but also obtained peak symmetry (expressed as Asymmetry factor: As) of selected benzodiazepines should be compared.
  3. It is better to use the lower limit of quantification (LLOQ) instead LOQ. Its value shouldn’t be lower than the lowest value of the standard curve.
  4. The authors claim that they validated the method according to the International Conference on Harmonization guidelines (ICH), however, they do not cite a proper reference.
  5. Add a section about extraction recovery. The authors wrote that the recovery values of selected BZDs were calculated by comparing the analyte peak areas obtained from plasma samples with those in standard methanolic solutions. Complete information is needed (for which concentrations, how many repetitions). Add obtained results to the discussion and conclusion section.
  6. Add information about the inter-day and intra-day accuracy. Add appropriate information to Table 2 (here use the term accuracy instead recovery).
  7. The authors did not test the developed method for real samples from patients treated with benzodiazepines? How do the authors know the method will work?
  8. There are some typos in the text. The text of the manuscript should be carefully checked.

Author Response

Ref.: Ms. No. analytica-1684739R1

Title: A simple and reliable liquid chromatographic method for simultaneous
determination of five benzodiazepine drugs in human plasma

Dear Prof. Dr. Marcello Locatelli 
Editor-in-Chief "Analytica"

Dear Ms. Andy Yang

Thank you so much for your hard efforts, constructive suggestions and comments on the manuscript. Those comments are fruitful for the improvement of the current work. We have carefully studied them and have tried our best to revise the manuscript according to your comments. We appreciate for your hard work earnestly, and we hope that the corrections will meet with your approval and the approval of editors of your distinguished journal. All corrections have been tracked through the revised manuscript. 

Sincerely yours,

Prof. Hassan M Albishri

Reviewers' comments:

First Review

  1. There are many methods of determining benzodiazepine drugs. The authors mentioned only some of them in the introduction. The presented results do not seem to be significantly new. The authors should clearly tell why their method is superior to the methods in the literature and should compare their results to the literature.

Response: Thank you so much. We appreciate for your hard work earnestly, and we hope that the current revised form will meet with your approval.A section 3.2.9 has been added in the revised text indicated the modifications in the current method compared to the previous published methods. As well, Table 4 has been inserted to show these modifications.

  1. Not only the column efficiency but also obtained peak symmetry (expressed as Asymmetry factor: As) of selected benzodiazepines should be compared.

Response: Thank you so much for you suggestions. As well, the comparison based on the peak symmetry among selected BZDs has been written in section 3.1.3.

  1. It is better to use the lower limit of quantification (LLOQ) instead LOQ. Its value shouldn’t be lower than the lowest value of the standard curve.

Response: Thanks so much. Based on your recommendation, the lower limit of quantification (LLOQ) has been used through the revised text instead LOQ.

  1. The authors claim that they validated the method according to the International Conference on Harmonization guidelines (ICH), however, they do not cite a proper reference.

Response: Thank you so much. Reference [17] demonstrating this point has been cited.

  1. Add a section about extraction recovery. The authors wrote that the recovery values of selected BZDs were calculated by comparing the analyte peak areas obtained from plasma samples with those in standard methanolic solutions. Complete information is needed (for which concentrations, how many repetitions). Add obtained results to the discussion and conclusion section.

Response: Thank you so much. A section about extraction recovery has been added (section 3.2.3). Further information about the concentrations and number of repetitions has been added in this section and conclusion.

  1. Add information about the inter-day and intra-day accuracy. Add appropriate information to Table 2 (here use the term accuracy instead recovery).

Response: Thank you so much. Appropriate information about the inter-day and intra-day accuracy has been added in section 3.2.6. and Table 2 (the term accuracy instead recovery).

  1. The authors did not test the developed method for real samples from patients treated with benzodiazepines? How do the authors know the method will work?

Response: Thanks so much. Recently, it has become very important in the institutions of forensic research and toxicology to know selective analytical methods for determination of BZDs in human plasma due to the complexity of matrix and the diversity of BZDs in the market having different and wide toxicity (overdose) ranges [9-13]. The forensic labs usually received the blood samples which were randomly withdrawn from humans e.g. drivers on the road and the response should be given within a few hours. Therefore, in the current work we focused on this demand and we developed a simultaneous analytical method for the screening and quantification of some BZDs, which commonly used in the market, within their therapeutic concentrations in the blood samples. The therapeutic ranges in blood of bromazepam, diazepam, clonazepam, lorazepam and nordaizepam are 80-170, 90-350, 100-400, 20-250 and 10-470 ng mL-1 [18]. The measurements of all studied BZDs therapeutic concentrations in plasma are in their ranges of linearity (10-500 ng mL-1). This information has been written in the revised manuscript.

  1. There are some typos in the text. The text of the manuscript should be carefully checked.

Response: Thank you so much for your fruitful suggestions. We did our best in order to carefully check the revised manuscript.

Please see the attachment for more details 

Reviewer 2 Report

The manuscript “A simple and reliable liquid chromatographic method for simultaneous determination of five benzodiazepine drugs in human plasma” proposes the use of a mobile phase with high content of pollutant organic solvent (methanol) after sample treatment of the sample involving liquid-liquid extraction with the same organic solvent.  

At first glance, in the presented manuscript there is not any significant novelty to add to the lots of applications reported on HPLC for the analysis of benzodiazepines (BZDs). Also in recent years, innovative methods based on surfactant-mediated mobile phases, involving direct injection without sample pre-treatment and using small amount of organic solvent or even avoiding its use, have been proposed. In fact, the authors claim as relevant novelty that the analysis of the selected mixture of BZDs was not found in previous publications. I do not see the convenience of repeating similar works with the unique difference of the selected group of solutes. I do not recommend publication in Analytica.  

Other comments: 

1) In the abstract, it is indicated that the proposed method is an alternative RPLC method. Alternative to what methods? 

2) Log Po/w of the studied compounds should be included.  

3) Analyses are carried out in spiked plasma samples, which is not realistic. Are the used concentrations within the physiological concentration range of the compounds in plasma?

 4) In section 3.1.2, it is indicated that the resolution improved at increasing amount of methanol. However, according to Fig.2, the resolution is better at decreasing amount of methanol. Acetonitrile was also used but the results are not even commented. How was the efficiency calculated? The effect of pH on retention and peak shape is related to the ionization character of solutes and the presence of residual silanols on the stationary phase, but explanations on this issue are quite superficial.  

5) Section 3.1.4. A figure would be needed. The effect of flow-rate is irrelevant.

 6) Section 3.2.3. LODs and LOQs should be specified for each solute.  

7) Blank plasma samples are not shown. 

Author Response

The manuscript “A simple and reliable liquid chromatographic method for simultaneous determination of five benzodiazepine drugs in human plasma” proposes the use of a mobile phase with high content of pollutant organic solvent (methanol) after sample treatment of the sample involving liquid-liquid extraction with the same organic solvent.  

At first glance, in the presented manuscript there is not any significant novelty to add to the lots of applications reported on HPLC for the analysis of benzodiazepines (BZDs). Also in recent years, innovative methods based on surfactant-mediated mobile phases, involving direct injection without sample pre-treatment and using small amount of organic solvent or even avoiding its use, have been proposed. In fact, the authors claim as relevant novelty that the analysis of the selected mixture of BZDs was not found in previous publications. I do not see the convenience of repeating similar works with the unique difference of the selected group of solutes. I do not recommend publication in Analytica.  

Response: Thank you so much. I appreciate for your hard work earnestly. A section 3.2.9 has been added in the revised text indicated the modifications in the current method compared to the previous published methods. As well, Table 4 has been inserted to show these modifications.

Other comments: 

1) In the abstract, it is indicated that the proposed method is an alternative RPLC method. Alternative to what methods?

Response: Thank you so much for your fruitful suggestions. Alternative word has been deleted in the abstract. 

2) Log Po/w of the studied compounds should be included.  

Response: Thanks so much. Log Po/w values of the studied compounds have been included in section 2.4.1.  

3) Analyses are carried out in spiked plasma samples, which is not realistic. Are the used concentrations within the physiological concentration range of the compounds in plasma?

Response: Thanks so much. Recently, it has become very important in the institutions of forensic research and toxicology to know selective analytical methods for determination of BZDs in human plasma due to the complexity of matrix and the diversity of BZDs in the market having different and wide toxicity (overdose) ranges [9-13]. The forensic labs usually received the blood samples which were randomly withdrawn from humans e.g. drivers on the road and the response should be given within a few hours. Therefore, in the current work we focused on this demand and we developed a simultaneous analytical method for the screening and quantification of some BZDs, which commonly used in the market, within their therapeutic concentrations in the blood samples. The therapeutic ranges in blood of bromazepam, diazepam, clonazepam, lorazepam and nordaizepam are 80-170, 90-350, 100-400, 20-250 and 10-470 ng mL-1 [18]. The measurements of all studied BZDs therapeutic concentrations in plasma are in their ranges of linearity (10-500 ng mL-1). This information has been written in the revised manuscript.

 4) In section 3.1.2, it is indicated that the resolution improved at increasing amount of methanol. However, according to Fig. 2, the resolution is better at decreasing amount of methanol. Acetonitrile was also used but the results are not even commented. How was the efficiency calculated? The effect of pH on retention and peak shape is related to the ionization character of solutes and the presence of residual silanols on the stationary phase, but explanations on this issue are quite superficial.

Response: Thank you so much for your fruitful comments. The explanation of the effect of methanol on the resolution among the studied drugs has been modified in the revised manuscript (section 3.1.2). As well, the effect of acetonitrile has been written in the same section. The calculation of column efficiency and the explanations about the effect of pH have been shown under section 3.1.3.

5) Section 3.1.4. A figure would be needed. The effect of flow-rate is irrelevant.

Response: Thank you so much. The explanation of effect of flow rate has been modified in the revised manuscript. Figure 4 has been presented in order to show the optimal chromatogram under optimal experimental conditions.

 6) Section 3.2.3. LODs and LOQs should be specified for each solute.

Response: Thank you so much. LODs and LOQs values have been specified for each solute in section 3.2.4.

7) Blank plasma samples are not shown. 

Response: Thank you so much for your fruitful suggestions. Fig. 5 has been modified with showing the chromatogram of blank plasma matrix. 

Please see the attachment for more details  

Reviewer 3 Report

Comments,

  1. Can you add brief information about previous methods for the determination of BZD analytes?
  2. What is the initial pH of the buffer? How did you adjust the pH of the mobile phase to 7?
  3. Did you run the individual standards to confirm the Retention time of analytes? It is better to add it as a supporting information.
  4. Please see the attached documents for the more comments.

Author Response

  1. Can you add brief information about previous methods for the determination of BZD analytes?

Response: Thank you so much for your fruitful suggestions. I appreciate for your hard work earnestly. A section 3.2.9 has been added in the revised text indicated the modifications in the current method compared to the previous published methods. As well, Table 4 has been inserted to show these modifications.

  1. What is the initial pH of the buffer? How did you adjust the pH of the mobile phase to 7?

Response: Thanks so much for your fruitful suggestions. You are right the initial pH of the buffer in the mobile phase is 7.0 which is very slightly altered with methanol addition and the mobile phase is still in the neutral pH range as described in the revised text.

  1. Did you run the individual standards to confirm the Retention time of analytes? It is better to add it as supporting information.

Response: Thank you so much. Yes, the individual standards of BZDs have been run in order to confirm the retention times of analytes. Supporting information has been added as S1 to S5 under section 3.2.1.   

  1. Please see the attached documents for the more comments.

Response: Thank you so much for your fruitful suggestions. All of your comments have been concerned and modified in the revised manuscript.

Round 2

Reviewer 1 Report

The developed method enables the simultaneous determination of selected benzodiazepines within their therapeutic concentration ranges. However, the analyzes were carried out only in spiked plasma samples. In real samples, apart from the investigational drugs, their metabolites will be present. How do the authors know that drugs will be separated from their metabolites? In the case of the application of DAD detection, this can be a significant problem in the determination of drugs.

The LLOQ is the lowest concentration at which the target analytes can be reliably measured and reported with a certain degree of confidence, which should be ≥ the lowest point in the calibration curve.

There are some typos in the text. The text of the manuscript should be carefully checked.

Author Response

Ref.: Ms. No. analytica-1684739R2

Title: A simple and reliable liquid chromatographic method for simultaneous
determination of five benzodiazepine drugs in human plasma

Dear Prof. Dr. Marcello Locatelli 
Editor-in-Chief "Analytica"

Dear Ms. Andy Yang

Thank you so much for your hard efforts, constructive suggestions and comments on the manuscript. Those comments are fruitful for the improvement of the current work. We have carefully studied them and have tried our best to revise the manuscript according to your comments. We appreciate for your hard work earnestly, and we hope that the corrections will meet with your approval and the approval of editors of your distinguished journal. As well, we did our best in order to carefully check the revised manuscript. All corrections have been tracked through the revised manuscript. 

Sincerely yours,

Prof. Hassan M Albishri

Reviewers' comments:

First Review

  1. The developed method enables the simultaneous determination of selected benzodiazepines within their therapeutic concentration ranges. However, the analyzes were carried out only in spiked plasma samples. In real samples, apart from the investigational drugs, their metabolites will be present. How do the authors know that drugs will be separated from their metabolites? In the case of the application of DAD detection, this can be a significant problem in the determination of drugs.

Response: Thank you so much. We appreciate for your hard work earnestly, and we hope that the current revised form will meet with your approval. As described in section 3.2.10, three concentrations 70, 200 and 400 ng mL-1 of standard analytes solutions were spiked to aliquots of human plasma solution. The recovery results of the plasma matrix for each concentration after comparing their areas of the peaks with areas of standard solutions peaks were ranged between 95.2-103.4% (intradays) and 90.5-108.8% (interdays) as shown in Table 2. Fig. 5 shows the chromatogram of a plasma matrix containing a mixture of 70 ng mL-1 of bromazepam, clonazepam, lorazepam, nordaizepam and diazepam in comparison with a blank plasma sample at the optimal conditions. This method has proven its reliability in the analysis of several BZDs including the most commonly used compounds in the market without any interference with other compounds in plasma matrix. Furthermore in literature, several methods using HPLC/DAD have been successfully applied to the separation and quantification of BZDS on C18 column in plasma samples without appearing any interference with their metabolites. Diazepam and its metabolites N-desmethyldiazepam were separated with high resolution in human plasma [28]. Clonazepam and its metabolites 7-aminoclonazepam were well separated: clonazepam at 34.380 min and 7-Aminoclonazepam at 12.924 [30]. Bromazepam and it's common metabolites  2-(2-amino-5-bromobenzoyl) pyridine were separated at retention times of 4.397, and 5.317 min, respectively [31]. Nordiazepam which is classified as diazepam metabolites from the effect of liver enzymatic behavior as mentioned in [28] was separated in the current method with a good resolution. All these information has been written in the revised manuscript.

  1. The LLOQ is the lowest concentration at which the target analytes can be reliably measured and reported with a certain degree of confidence, which should be ≥ the lowest point in the calibration curve.

Response: Thank you so much for your suggestions. Based on your recommendation, the lower limit of quantification (LLOQ) values have been cited in section 3.2.4. Moreover, the linearity ranges of selected analytes have been modified using extrapolation in the calibration curves; Their values have been cited under section 3.2.2 and in conclusion.

  1. There are some typos in the text. The text of the manuscript should be carefully checked.

Response: Thank you so much for your fruitful suggestions. We did our best in order to carefully check the revised manuscript. Moreover, Man with a native English language has checked the revised manuscript.

Reviewer 2 Report

As I previously commented, this manuscript has a low quality based on lack of novelty. The selection of a concrete group of BZDs does not fill a gap that needs to be revisited or clarified. 

Author Response

Second Review

As I previously commented, this manuscript has a low quality based on lack of novelty. The selection of a concrete group of BZDs does not fill a gap that needs to be revisited or clarified. 

Response: Thank you so much. We appreciate for your hard work earnestly, and we hope that the current revised form will meet with your approval. Section 3.2.9 has been added in the revised text in order to indicate the useful modifications in the current method in a comparison with the previous published methods. As well, Table 4 has been inserted to briefly show this comparison. In order to overcome the difficult and multi steps extraction procedures, this work developed a simple, reliable, and accurate liquid-liquid extraction followed by a simple RPLC with LOD values in plasma matrixbetween 1.26-2.09 ng mL-1 that overcome most of the previous methods. As we can observe in the comparison table, several published papers have concerned with the quantification of a concrete group including 1,2,3,4 or 5 number BZDs which is lower than or equal the number of BDZs in the current work. Furthermore, the current method has proven its reliability in the analysis of the most commonly used BZDs in the market without any interference with their metabolites or other compounds in plasma matrix. Moreover, the measurements of all BZDs therapeutic concentrations in plasma are in their ranges of linearity (1-500 ng mL-1). The therapeutic ranges in blood of bromazepam, diazepam, clonazepam, lorazepam and nordaizepam are 80-170, 90-350, 100-400, 20-250 and 10-470 ng mL-1. More work, that we already started, has focusing on the modification of the current work with greener mobile phases including the quantification of larger window of BZDs compounds. All these information has been written in the revised manuscript.

Reviewer 3 Report

Thanks to addressing all of my comments for improving the manuscript for the publication.

Author Response

Thank you so much for your fruitful suggestions. I appreciate your hard work earnestly.

Please see the attachment for the last version of the manuscript. 

Round 3

Reviewer 1 Report

The manuscript can be accepted for publication in Analytica.

Reviewer 2 Report

My opinion has been given to Editors